# Long-Term Exposure to Fine and Coarse Particulate Matter and COVID-19 Incidence and Mortality Rate in Chile during 2020

**DOI:** 10.3390/ijerph18147409

**Published:** 2021-07-11

**Authors:** Macarena Valdés Salgado, Pamela Smith, Mariel A. Opazo, Nicolás Huneeus

**Affiliations:** 1Center for Climate and Resilience Research CR2, FONDAP N°15110009, Santiago 8370449, Chile; macavaldes@uchile.cl (M.V.S.); mariel.opazod@gmail.com (M.A.O.); nhuneeus@dgf.uchile.cl (N.H.); 2Programa de Epidemiología, Escuela de Salud Pública, Facultad de Medicina, Universidad de Chile, Santiago 8380453, Chile; 3Red de la Pobreza Energética—RedPE, Santiago 8320000, Chile; 4Departamento de Geografía, Universidad de Chile, Santiago 8331051, Chile; 5Project FONDECYT-INITIATION N°11180990 Social Construction of the Urban Climate: towards Quality and Climate Justice in Chilean Cities, Santiago 8320000, Chile; 6Chile Departamento de Geofísica, Facultad de Cs Físicas y Matemáticas, Universidad de Chile, Santiago 8370449, Chile

**Keywords:** COVID-19, SARS-CoV-2, air pollution, climate, South America, environmental indicators

## Abstract

Background: Several countries have documented the relationship between long-term exposure to air pollutants and epidemiological indicators of the COVID-19 pandemic, such as incidence and mortality. This study aims to explore the association between air pollutants, such as PM_2.5_ and PM_10_, and the incidence and mortality rates of COVID-19 during 2020. Methods: The incidence and mortality rates were estimated using the COVID-19 cases and deaths from the Chilean Ministry of Science, and the population size was obtained from the Chilean Institute of Statistics. A chemistry transport model was used to estimate the annual mean surface concentration of PM_2.5_ and PM_10_ in a period before the current pandemic. Negative binomial regressions were used to associate the epidemiological information with pollutant concentrations while considering demographic and social confounders. Results: For each microgram per cubic meter, the incidence rate increased by 1.3% regarding PM_2.5_ and 0.9% regarding PM_10_. There was no statistically significant relationship between the COVID-19 mortality rate and PM_2.5_ or PM_10_. Conclusions: The adjusted regression models showed that the COVID-19 incidence rate was significantly associated with chronic exposure to PM_2.5_ and PM_10_, even after adjusting for other variables.

## 1. Background

Since the first COVID-19 case in Wuhan, China, the pandemic caused by SARS-CoV-2 has caused more than 162 million cases and 3.3 million deaths globally. According to official records, the first case of COVID-19 in Chile was reported on 3 March 2020. Since then, by May 2021, more than 1,500,000 cases and around 172,000 deaths occurred in Chile [1].

A wide variety of manifestations have been reported among people with COVID-19, ranging from asymptomatic cases to mild symptoms and even severe illness and death [2]. Epidemiological indicators, such as incidence, mortality, or case fatality rate, vary among countries [3]. This variability has motivated research on risk factors from the individual to the environmental level.

As mentioned in previous studies, the risk factors for severe COVID-19 illness are: age, sex (male), smoking, sedentariness, comorbidities such as diabetes and obesity, low oxygen saturation at hospital admission, or receiving mechanical ventilation during hospitalization [4,5,6]. These studies have pointed out that any condition that promotes an inflammatory response could exacerbate the effects of the coronavirus. Exposure to air pollutants increases the probability of contracting infectious diseases due to their oxidative effect, which alters the immune response and the permeability of cells to viral entry [7,8]. This increases the morbidity and mortality from respiratory and cardiovascular diseases in places with high levels of pollution [9,10,11].

Regarding the COVID-19 pandemic, some ecological studies have shown a relationship between air pollution and incidence and mortality rate [12,13]. In the USA, municipalities with greater chronic exposure to fine particulate matter (particular matter with a diameter less than or equal to 2.5 μm, or PM_2.5_) had a 2 to 15% higher risk of dying from COVID-19 for every microgram per cubic meter increase in this pollutant [14]. In London, nitrogen oxides (NO_X_) were positively correlated with COVID-19 cases and deaths [15]. In 63 Chinese cities, the basic reproductive number was associated with the daily concentration of nitrogen dioxide (NO_2_) [16]. In northern Italy, for every microgram per cubic meter increase in chronic exposure (considered the annual average between 2015 and 2019) to PM_2.5_, the risk of death from COVID-19 increased by 9% [17]. A time-series study conducted in India showed that pollutants such as sulfur dioxide (SO_2_), carbon monoxide (CO), NO_2_, and PM_2.5_ were related to daily cases of COVID-19 [18]. In China, short-term exposure (from 0 to 14 days) to air pollutants, such as particulate matter with a diameter less than or equal to 10 μm (PM_10_), PM_2.5_, CO, NO_2_, and ozone (O_3_), was associated with a larger number of confirmed COVID-19 cases [19].

Concerning other environmental factors and SARS-CoV-2 transmission, this novel coronavirus seems to be easily transmissible in zones where temperatures oscillate between 5 and 15 °C [20]. No conclusive evidence exists to establish that relative humidity, precipitation, or cloud cover are related to SARS-CoV-2 transmission [20]. The first study conducted in Chile focused on COVID-19 and climate, and it divided Chile into seven climatic zones with data from February to March 2020 [21] (Correa-Araneda et al., 2020). The authors found that both mean temperature and relative humidity had a negative effect on the absolute rate of COVID-19 infection. The authors have highlighted the relevance of adding other variables (for instance, indicators of human behavior or indoor conditions) to fully explain the transmission of COVID-19. In this context, social vulnerability, represented by housing conditions, could also contribute to the variability of COVID-19 indicators. Indeed, the World Health Organization has pointed out that infrastructure, size, and overcrowding are important variables to consider in the transmission of infectious diseases such as COVID-19 [22].

This pandemic is expected to continue to be a significant challenge given its impact in 2020 [23]. It is pertinent not only to identify clinical factors related to the disease in individuals, but also to identify environmental factors related to the incidence and mortality in the population. The objective of this research is to study the association between chronic exposure to PM_2.5_ and PM_10_ and health indicators of COVID-19, such as the incidence and mortality rate, at the communal level while taking into account the population, housing, and climate characteristics of each commune. The study’s rationale is to provide local evidence to health authorities, politicians, and society about the effects of air pollution on the current pandemic.

## 2. Materials and Methods

### 2.1. Study Design and Population

Continental Chile is located on the southwestern coast of South America, extending from 17° S to 56° S between the Andes Mountain range and the Pacific Ocean. Administratively, Chile is divided into 16 regions and 346 communes. This study used data from 188 communes based on their availability: 6 communes from the northern macrozone (Regions XV, I, II, III, IV); 113 from the central zone (Regions V, XIII, VI, VII); and 69 from the southern zone (Regions XVI, VIII, IX, X, XIV, XI, XII). This study was an ecological design as our units of analysis were defined geographically rather than by individuals, comparing COVID-19 incidence and mortality rate among Chilean communes in 2020.

### 2.2. Variables and Sources of Information

#### 2.2.1. Health Variables

The confirmed COVID-19 cases and deaths were obtained at the commune level from the website of the Chilean Ministry of Science [24]. A case of SARS-CoV-2 infection is confirmed when a person returns a positive nucleic acid amplification test, or has a positive SARS-CoV-2 antigen test and meets either the probable case definition or is asymptomatic and had contact with a probable or confirmed case. The ICD codes used to define a COVID-19 death were U07.1 and U07.2.

For each commune, the incidence and mortality rate were estimated from the confirmed cases and deaths that occurred in 2020 normalized by their population. The population numbers were obtained from the National Statistics Institute of Chile based on the 2017 Census [25].

#### 2.2.2. Air Pollution

The annual average concentration of PM_2.5_ and PM_10_ was estimated through the WRF-CHIMERE modeling system for each commune [26]. It simulates the physical–chemical processes of the different atmospheric pollutants based on emissions from the main anthropogenic sectors: residential, transportation, industry, and energy. This modeling system has been successfully applied to simulate the dispersion of atmospheric pollutants in central and southern Chile as well as the exposure to particulate matter of public transport users [27,28,29].

The PM_2.5_ and PM_10_ concentrations were estimated for the year 2016 along continental Chile at a 10 km spatial grid that was then interpolated to a resolution of 2 km. Contrary to the health data considered at the communal level, the annual averages for PM_2.5_ and PM_10_ concentrations were considered at a city level. Communes are a minor and basic administrative division of the whole territory; they are not restricted to urban limits and are in general a combination of urban and rural areas. In communes with large rural or unpopulated areas, large differences in concentrations exist between the communal and city average (Figure 1, see case of Coyhaique II), with the latter being higher and closer to the observations. Therefore, since using the communal average for this study could potentially underestimate the exposure of the population to these pollutants and consequently their health impact, and considering only 12.2% of the population lives in a rural area at a national level according to the national 2017 Census, the city average was used to calculate the concentration of particulate matter instead.

The annual average PM_2.5_ and PM_10_ concentration was estimated as the average of daily concentrations for the year 2016 of all grid points corresponding to the city. The impacts on anthropogenic emissions due to the restrictions applied during the COVID-19 pandemic have not yet been quantified and therefore the year 2020 could not be simulated for the present study. Therefore, a different year needed to be used and the year 2016 represents the best compromise between available observations and the absence of large emission events such as forest fires.

The Chilean air quality (AQ) monitoring network (https://sinca.mma.gob.cl, accessed on 24 May 2021) has a total of 28 stations with PM_2.__5_ and PM_10_ data available for the year 2016. The location of the stations varies from completely urban to small isolated cities, and their latitudes range between 45° S and 33° S. The annual PM_2.__5_ concentrations for the available stations range from 11 to 67 µg/m^3^, being higher for the higher latitudes where firewood heating is predominantly used and where maximum daily values can reach up to 500 μg/m^3^. For PM_10_, the observed concentrations range from 20 to 87 µg/m^3^ and maximum daily averages reach up to 530 µg/m^3^. Contrary to PM_2.__5_ which, in general, shows a gradient of increasing concentrations with increasing latitude, PM_10_ does not present the same variability, suggesting different emission profiles among cities in central and southern Chile.

To assess the performance of the model to simulate the observations, a comparison between the measured and simulated yearly daily average concentrations of PM_2.__5_ and PM_10_ was conducted for a selected number of stations. This comparison reveals that the measured concentrations are, in general, reproduced when compared against the simulated observation (Figure 1). However, except for Osorno, a persistent underestimation of the modeled city concentration against the stations’ observations can be noticed. We highlight that the observed value corresponds to the measured concentration at a given point and therefore reflects the air quality at that given location and its surroundings, whereas the city average represents the average of simulated concentrations over an area. Therefore, a persistent underestimation of the measured yearly average by the simulated city yearly average does not suggest difficulties of the model to reproduce the observations, but rather the fact that both have different spatial representativity. Despite this limitation, the model approach was chosen given that the AQ network has a limited number of stations in a select group of cities, thus providing only partial coverage of the air quality in the country. The model provides AQ information for communes not covered by the Chilean monitoring network, allowing the application of the study to a larger fraction of the Chilean population.

#### 2.2.3. Confounders

For each commune, we considered potential confounders, such as:Temperature and humidity: The average annual air temperature and relative humidity from 1980–2010 were obtained from ARClim, which is an open-source meteorological dataset [30]. This dataset was chosen due to its official endorsement given by the Chilean Ministry of Environment, and because its spatial resolution (5 × 5 km) provides data for all 188 communes.Elderly index: Proportion of the elderly population (65 years old or more) for 2020 based on projections of the 2017 Census.Self-report of health: Proportion of the population that have declared absence of chronic or acute diseases, based on the National Socioeconomic Characterization Survey [31]. The reported illnesses include hypertension, diabetes, myocardial infarction, stroke, chronic obstructive pulmonary disease, asthma, thyroid disease, cancer, and lupus.Percentage of houses with moderate overcrowding: Number of houses with moderate overcrowding out of the total number of houses in each commune. The overcrowding corresponds to the ratio of the number of people to the number of bedrooms. Moderate overcrowding is defined as 2.5 to 4.9 people per bedroom [32].Percentage of houses built before 2000: Number of houses built before 2000 out of each commune’s total number of houses. Homes built before 2000 were not subject to the thermal standard considered in the first stage of regulations associated with thermal insulation of housing [32].Mobility index: Sum of the internal and external mobility indexes. The internal index is a measure of the number of trips that occur within a commune, whereas the latter corresponds to the number of trips made outside of the commune. Both indexes were summed and normalized by the population of the administrative unit [33].

### 2.3. Statistical Analyses

Mean, standard deviations, and range describing the health and socio-environmental characteristics of the communes are illustrated in Figure 2 and Table 1. Incidence and mortality rate according to the level of exposure to PM_2.5_ and PM_10_ are depicted in Figure 3 and Figure 4, respectively. The effect of chronic exposure to PM_2.5_ and PM_10_ on COVID-19 incidence and mortality rate was estimated using negative binomial regression, a generalization of Poisson regression, which allows consideration of the overdispersion of incidence and mortality rate data. In the following equation, *µ_i_* represents the mean incidence (or mortality) rate in a period of time *t_i_*, *x_k_* represents the *k* predictors included in the model, and ε represents the overdispersion
$${µ}_{i}=\exp\;(\ln(t_{i})+\beta_{1}{x_{1}}_{i}+\beta_{2}{x_{2}}_{i}+\cdots +\beta_{k}{x_{k}}_{\;}+\varepsilon$$

Therefore, to study the association between PM_2.5_ and PM_10_ on COVID-19 incidence and mortality rate at the communal level, we estimated crude and adjusted models that included potential confounders based on previous studies, such as aging of the population (whose proxy is elderly index), the health status of the population (whose proxy is self-reported health), degree of overcrowding in the commune (whose proxy is percentage of houses with moderate overcrowding), ventilation (whose proxy is percentage of houses built before 2000), mobility during the pandemic (whose proxy is mobility index in each commune in 2020), and the long-term exposure to meteorological factors (average of temperature and relative humidity).

Maps were made using ArcGIS 10.6 software and the statistical analyses using Stata version 14 statistical software.

## 3. Results

Nationally, the incidence and mortality rates in 2020 were 4574 cases per 100,000 habitants and 84 deaths per 100,000 habitants, respectively. Although the yearly PM_2.5_ and PM_10_ concentration averaged over all communes is below the Chilean and the World Health Organization (WHO) air quality standards, the maximum annual concentration for both pollutants revealed that there are communes that significantly exceed both air quality standards (Table 1). Demographically, the Chilean population is old, with a high percentage of elderly (above 10%). Socially, some communes have a high proportion of houses with moderate overcrowding and without thermal standards (built before 2000). Climatically, the temperature and humidity indicate that both warm and dry climates are present throughout the studied territory (Table 1).

Most communes register incidences over 2875, but in the southern Araucanía Region, some communes exceed 6200 infections per 100,000 inhabitants (Figure 2A), spatially coinciding with the highest recorded mortality values (Figure 2B). Chronic exposure to PM_10_ (Figure 2C) and PM_2.5_ (Figure 2D) varies throughout the Chilean territory. In both cases, communes exceeding the maximum annual concentration value defined by the WHO (PM_10_: annual average 20 µg/m^3^; PM_2.5_: 10 µg/m^3^) are shown in yellow, and those in red also exceed the Chilean AQ standard (PM_10_: annual average 50 µg/m^3^; PM_2.5_: 20 µg/m^3^). In general, there are high concentration values of particulate matter of both fractions and a high incidence of COVID-19 in the central and south-central Chilean communes.

There is spatial coincidence between high average overcrowding (average 2.5 to 4.9 people per room—Figure 2I) and incidence and mortality due to COVID-19, particularly in the south-central and the far northern Chilean communes. This spatial coincidence is also present when looking at the proportion of houses built before 2000, that is, before the implementation of the housing norm (Figure 2H); since they were built without thermal insulation requirements, they are more exposed to the cold and therefore need more heating, which in south and south-central Chile involves the use of firewood with the consequent impact on indoor pollution.

Bivariate analysis between COVID-19 incidence and mortality rate and chronic exposure to PM_2.5_ showed positive correlations, which means that those communes with the highest levels of chronic exposure to PM_2.5_ showed the highest incidence and mortality rates (Figure 3); the same tendencies were observed for incidence and mortality rates and chronic exposure to PM_10_ (Figure 4).

Crude models showed that the associations between COVID-19 incidence rate and PM_2.5_ were statistically significant. Multivariate models confirmed these associations even after adjusting by confounders. For each microgram per cubic meter, the incidence rate increased by 1.2% regarding PM_2.5_ and 0.9% regarding PM_10_. Other variables were also associated with the COVID-19 incidence rate. For example, the percentage of houses with moderate overcrowding and relative humidity were directly associated with incidence, whereas the elderly index was inversely associated with this outcome (Table 2).

Even though crude models showed an association between air pollutants and mortality rate, after adjusting by confounders, this association was no longer statistically significant (Table 3).

## 4. Discussion

This study’s results are partially consistent with those shown in the US, Italy, and China. In our data, the chronic exposure to particulate matter was associated with COVID-19 incidence but not mortality rate. Previous studies, such as Wu’s article on counties from the United States, revealed that the risk of dying from COVID-19 increased by 8% for each additional unit of PM_2.5_ [14]; in Coke’s study conducted in northern Italian communes, the risk increased by 9% [17]; and in Cole’s study in the Netherlands, for each unit increase in PM_2.5_, 2.3 more deaths were recorded [34]. In our crude models, we found an association between chronic exposures to particulate matter, coarse and fine, and mortality rate. However, after adjusting by other covariates, this association was no longer statistically significant. We hypothesize that a greater number of communes is required to improve the statistical power to be able to confirm these results. Furthermore, the average concentrations of PM_2.5_ and PM_10_ in this study were slightly lower than in the previously mentioned studies, which could potentially explain our differences.

Regarding the associations observed between PM_2.5_ and PM_10_ and the incidence rate of COVID-19, despite the fact that our results have shown a higher incidence rate in those territories with a higher average concentration of PM_2.5_ or PM_10_, the estimates were modest compared to previous studies [13]. Some of the studies described in Bhaskar’s review have mentioned an average exposure to air pollutants higher than ours; perhaps we could have found association measures (i.e., incidence rate ratios) close to those studies if we had had higher concentrations of air pollutants.

Within the several studies carried out during the epidemics of SARS in 2003 and H1N1 in 2017, chronic exposure to air pollutants was associated with health indicators such as incidence and mortality. This consistency has been highlighted in the editorial letter of Contini and Costabile [35], suggesting that chronic exposure to air pollution could shape the results of the current pandemic.

Short-term exposure to air pollutants is also highlighted in other studies, showing a positive correlation with incidence and mortality indicators [18,19]. As Bhaskar indicated in his review [13], long- and short-term exposures to air pollutants induce oxidative stress and decrease cellular immunity and chemical response. Additionally, the expression of the angiotensin-2 converting enzyme, which is involved in the virus’s entry into lung cells, is associated with chronic exposure to air pollutants [8,36].

In the present study, we found a direct association between COVID-19 incidence rate and relative humidity, an inverse association between COVID-19 mortality rate and relative humidity, and a direct association between COVID-19 mortality rate and temperature. On the other hand, Correa-Araneda’s study, which was carried out with data from 121 Chilean cities, showed an inverse association between the infection rate of COVID-19 and the mean temperature and relative humidity while taking into account wind speed [21]. Unfortunately, we do not have the wind speed available for all cities. If it were available, comparisons with other studies would have been interesting to incorporate into this research. In the study by Pramanik et al. in Russia, the humidity was not directly correlated with COVID-19 cases; however, the authors pointed out that this relationship depends on the climate type [37]. In India, Gupta’s study has emphasized that climatic factors do not explain the variability of the number of infected people [38]. As can be seen, there is no conclusive evidence regarding health indicators of COVID-19 and meteorological factors yet.

Social distancing and masks are measures recommended by the WHO to reduce the risk of spreading the virus. Our results are consistent with these measures, finding that the average overcrowding in homes is a factor that determines higher incidences of COVID-19. Patel’s article highlighted this issue, in which he describes the difficulty of adhering to certain sanitary measures, such as quarantine, in cities that are highly populated and people live in a way that they suffer the consequences of this pandemic with greater intensity [39]. Taking into account the growing evidence on the airborne transmission of SARS-CoV-2, especially in closed spaces, the use of masks in overcrowded environments could be the main way to avoid outbreaks [40].

Regarding our strengths and limitations, this is the first study in Chile that seeks to find an association between COVID-19 and long-term exposure to pollution, while adjusting for potential social and climatic confounders. Even when several confounders were included, residual confounding was possible. Likewise, the data on climatic parameters correspond to the 30-year average and likely differ from the specific conditions observed during 2020. Like all ecological studies, these results do not establish relationships at the individual level. Chile has 346 communes, and our analyses were based on 188; if we had data regarding air pollutants for the total number of communes, perhaps we could have reduced random error and the potential selection bias and achieved better statistical power.

## 5. Conclusions

Chronic exposure to PM_2.5_ and PM_10_ was associated with a higher risk of COVID-19 incidence at the communal level in Chile. It is noteworthy that variables such as household overcrowding, housing quality, humidity, and temperature also shaped our epidemiological indicators.

From an environmental health perspective, our results strengthen the previously published evidence regarding long-term exposure to air pollutants—such as particulate matter—associated with worse COVID-19 epidemiological indicators. Therefore, policies and other measures focused on air pollution mitigation will protect the population and aid in overcoming future respiratory epidemics.

Futures studies need to go down to the individual level to consider the characteristics of people as well as environmental factors.

## Figures and Tables

**Figure 1 ijerph-18-07409-f001:**
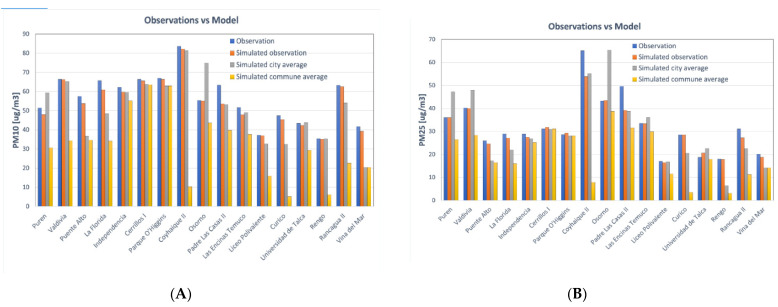
WRF-CHIMERE modeling system to estimate air pollutants. (**A**) Comparison between PM_10_ estimations based on model and monitoring based on stations. (**B**) Comparison between PM_2.5_ estimations based on model and monitoring based on stations. Yellow: simulated commune yearly average; gray: simulated city yearly average; orange: yearly average of simulated observation (closest point to the station); blue: yearly average of station measurements.

**Figure 2 ijerph-18-07409-f002:**
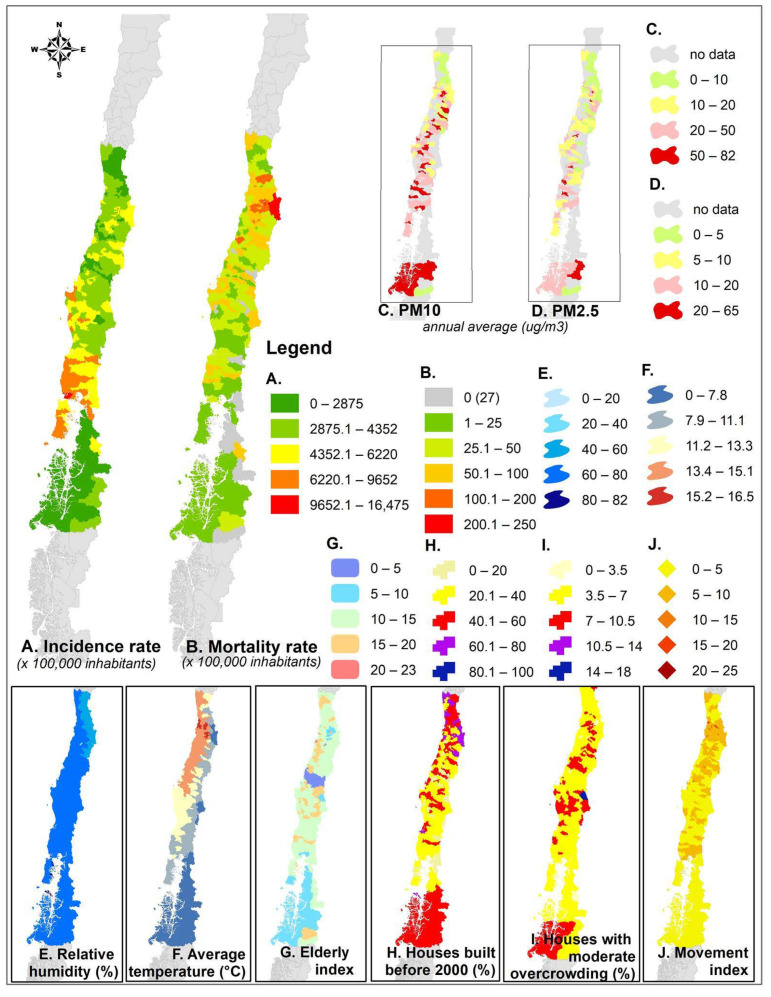
Spatial distribution of incidence and mortality rate, PM_10_, PM_2.5_, relative humidity, annual average temperature, elderly index, % houses built before 2000, % houses with moderate overcrowding, and mobility index.(**A**) Incidence of COVID-19 per 100,000 inhabitants during 2020 (**B**) Mortality of COVID-19 per 100,000 inhabitants during 2020; (**C**) Annual average of PM_10_ concentration; (**D**) Annual average of PM_5_ concentration; (**E**) Annual average of relative humidity; (**F**) Annual average of air temperature; (**G**) Elderly index; (**H**) Percentage of houses built before 2000 year; (**I**) Percentage of houses with moderate overcrowding; (**J**) Movement index.

**Figure 3 ijerph-18-07409-f003:**
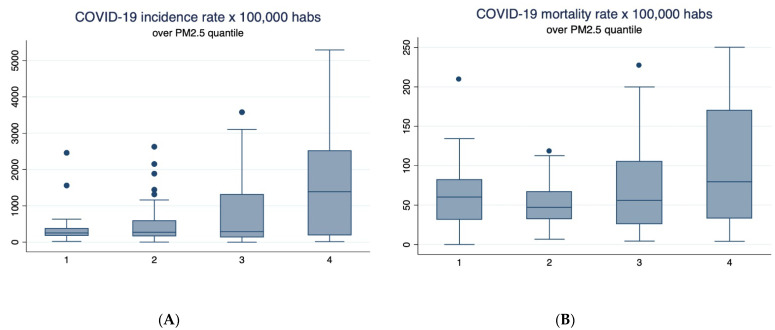
Box plot of COVID-19 incidence and mortality rate above quantile of PM_2.5_. (**A**) Relationship between quartiles of PM_2.5_ and COVID-19 incidence rate × 100,000. (**B**) Relationship between quartiles of PM_2.5_ and COVID-19 mortality rate × 100,000. The quartiles of PM_2.5_ are 1: [2.06–7.46 µg/m^3^); 2: [7.46–14.12 µg/m^3^); 3: [14.12–22.60 µg/m^3^); 4: [22.60–65.43 µg/m^3^].

**Figure 4 ijerph-18-07409-f004:**
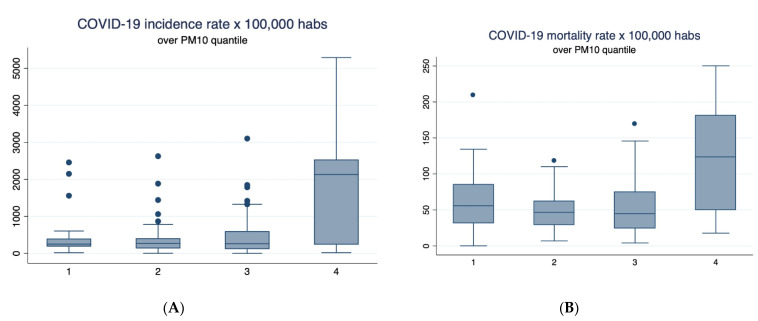
Box plot of COVID-19 incidence and mortality rate above quantile of PM_10_. (**A**) Relationship between quartiles of PM_10_ and COVID-19 incidence rate × 100,000. (**B**) Relationship between quartiles of PM_10_ and COVID-19 mortality rate × 100,000. The quartiles of PM_10_ are 1: [3.43–11.31 µg/m^3^); 2: [11.31–18.30 µg/m^3^); 3: [18.30–32.61 µg/m^3^); 4: [32.61–81.43 µg/m^3^].

**Table 1 ijerph-18-07409-t001:** Health and socio-environmental statistics in 188 Chilean communes, 2020.

	*n*	Mean	St. Deviation	Min	Max
Incidence × 100,000 inhabitants	188	4501	1423	1786	9071
Mortality × 100,000 inhabitants	188	71	56	0	250
Annual average PM_2.5_, μg/m^3^	188	16	10	2	65
Annual average PM_10_, μg/m^3^	188	24	18	3	81
Elderly index, %	188	13	3	4	23
Self-reported health, %	188	73	4	57	85
Houses with moderate overcrowding, %	188	6	2	1	12
Houses built before 2000, %	188	45	16	11	88
Mobility index	188	7	3	0	30
Average temperature, °C	188	13	2	5	16
Relative humidity, %	188	67	7	48	79

**Table 2 ijerph-18-07409-t002:** Crude and adjusted association between COVID-19 incidence rate and PM_2.5_ and PM_10_ in 188 Chilean communes in 2020.

	Crude Models			Adjusted Models		
	**IRR**	**95%CI**	***p*** **-Value**	**IRR**	**95%CI**	***p*** **-Value**
Annual average PM_2.5_, μg/m^3^	1.015	1.011–1.019	<0.001	1.012	1.008–1.017	<0.001
Elderly index, %				0.989	0.975–1.004	0.155
Self-reported health, %				1.007	0.998–1.016	0.115
Houses with moderate overcrowding, %				1.043	1.018–1.068	0.001
Houses built before 2000, %				1.000	0.998–1.003	0.769
Mobility index				0.997	0.982–1.012	0.666
Average temperature, °C				0.997	0.977–1.017	0.742
Relative humidity, %				1.012	1.006–1.018	<0.001
Annual average PM_10_, μg/m^3^	1.009	1.006–1.011	<0.001	1.009	1.007–1.011	<0.001
Elderly index, %				0.985	0.972–0.999	0.041
Self-reported health, %				1.003	0.995–1.012	0.472
Houses with moderate overcrowding, %				1.038	1.015–1.063	0.001
Houses built before 2000, %				0.999	0.997–1.002	0.627
Mobility index				0.998	0.983–1.012	0.759
Average temperature, °C				0.988	0.969–1.007	0.224
Relative humidity, %				1.015	1.009	<0.001

IRR: incidence rate ratio; 95%CI: 95% confidence interval.

**Table 3 ijerph-18-07409-t003:** Crude and adjusted association between COVID-19 mortality rate and PM_2.5_ and PM_10_ in 188 Chilean communes in 2020.

	Crude Models			Adjusted Models		
	**IRR**	**95%CI**	***p*-Value**	**IRR**	**95%CI**	***p*-Value**
Annual average PM_2.5_, μg/m^3^	1.022	1.011–1.033	<0.001	1.004	0.996–1.011	0.361
Elderly index, %				0.984	0.956–1.013	0.271
Self-reported health, %				0.998	0.980–1.016	0.834
Houses with moderate overcrowding, %				1.035	0.986–1.086	0.161
Houses built before 2000, %				1.011	1.005–1.016	<0.001
Mobility index				1.004	0.969–1.039	0.842
Average temperature, °C				1.154	1.111–1.198	<0.001
Relative humidity, %				0.958	0.946–0.970	<0.001
Annual average PM_10_, μg/m^3^	1.018	1.012–1.023	<0.001	1.003	0.999–1.008	0.144
Elderly index, %				0.982	0.954–1.011	0.221
Self-reported health, %				0.997	0.979–1.015	0.707
Houses with moderate overcrowding, %				1.031	0.983–1.082	0.210
Houses built before 2000, %				1.010	1.005–1.016	<0.001
Mobility index				1.003	0.969–1.038	0.860
Average temperature, °C				1.149	1.106.1.194	<0.001
Relative humidity, %				0.959	0.947–0.970	<0.001

IRR: incidence rate ratio; 95%CI: 95% confidence interval.

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
