# Peer review of "Long-Term Exposure to Fine and Coarse Particulate Matter and COVID-19 Incidence and Mortality Rate in Chile during 2020"

_ijerph, 2021, doi:10.3390/ijerph18147409_

Round 1

Reviewer 1 Report

In part 2.2.3. Air pollution the authors mention annual mean values on PM2.5 and PM10 concentrations that were derived from the model. Are there measured PM concentrations with reference method? And how different are model predictions from PM measured concentrations? 

Author Response

Comment 1: “In part 2.2.3. Air pollution the authors mention annual mean values on PM2.5 and PM10 concentrations that were derived from the model. Are there measured PM concentrations with reference method? And how different are model predictions from PM measured concentrations?”
Answer: We have added more details in this subsection, which is currently called
“2.2.2. Air pollution” on pages 7-9. Also, we add a new figure (figure1) to illustrate the estimations of our WRF-CHIMERE models with the reference measurements. We also have improved the description of our methods and results to correct the errors noted by the referee. Additionally, a native English speaker did the fine/minor English editions required by the referee.

Reviewer 2 Report

Int. J. Environ. Res. Public Health

Title: Long-term exposure to fine and coarse particulate matter and COVID-19 incidence and mortality rate in Chile during 2020

In this study the authors investigate the association between particulate matter (PM2.5 and PM10) and COVID-19 incidence and mortality rates during 2020 in Chile. It is a long-term epidemiological study, with a design of an ecological study. In fact, there is some evidence that SARS-CoV-2 can exist on particulate matter, suggesting that air pollution can contribute to viral spread. From this point of view there is the rational for carrying out long-term and short-term studies specifically aimed at confirming or excluding the presence of the SARS-CoV-2 and its potential virulence on particulate matter in different cities around the world. To my knowledge, there have been only few studies conducted in South America on this topic, this would speak in favor of this study after consideration of comments that the authors might address in a revision (not in order of importance).

2.2.2: Health variables

” The COVID-19 confirmed cases and deaths were obtained at commune level from the website of the Chilean Ministry of Science” – there is no information how the COVOD-19 cases were confirmed, by PCR? ICD code?. Please provide the information

2.2.3: Air pollution

The annual average concentration of PM2.5 and PM10 were estimated by a chemical transport model. It is not clear for which year the PM concentration were modeled. Please state it. How was the model validated? I guess that in Chile network monitoring stations are operated. It would be nice to see the comparison of the modeled and measured PM2.5 and PM10 values.

I don’t understand the sentence “For each commune, the annual average of PM2.5 and PM10 was estimated as the average of the daily concentrations estimated by the model for each grid point”: the annual average was estimated as the average of daily concentrations? Please clarify it.

2.2.4. Temperature and humidity

The authors state that the average temperature and relative humidity were calculated based on data from 1980 and 2010. I agree that this is a long time period, however we know that we are currently facing a climate change and that the last few years are not necessarily comparable to the years before 2010. The last few years have been globally much warmer than the years before. How sure could the authors be that the year 2020 can be well represented by the mean values for the period 1980-2010?

2.2.5. Confounders

Self-report of health: this could be described in more detail. Which diseases are they, at least which groups of diseases are included?

Movement index: please provide the definition of “movement index”

  1. Results

Table 1: Obviously the results of the air pollution are available only for 188 communes. What is the reason for it? I guess that the analysis was done only for the 188 communes with the PM2.5 and PM10 modeled values. Please restrict all data (in tables and figures) only to those communes, or at least provide a second table with the health and socio-environmental statistics only for n=188 (and not 346). The same is valid for Figure 1 – please show the spatial distribution of all health variables, temperature and humidity as well as of the confounders only for the communes included into the analysis.

Figure 2 and figure 3: please state the values of the quantiles for PM2.5 and PM10.

Table 2 and 3: Please explain the abbreviations IRR and IC95%. Please show the values of the confidence intervals (I assume that “IC95%” means this) in brackets and separated by something (colon or dash), but not in two different columns (for example (1.008 - 1.017)).

Why are the statistically significant IRR values in Table 2 in bold, but this is not done in Table 3?

“Adjusted models confirmed these associations, taking into account important confounders. For each microgram per cubic meter, the incidence rate increased by 1.3% regarding PM2.5 and 0.9% regarding PM10”: I see in Table 2 increase of the incidence rate by 1.2% regarding PM2.5 (not 1.3%).

“For each microgram per cubic meter, the mortality rate increased by 0.8% regarding PM2.5 and 0.6% regarding PM10 (Table 2 and 3)”: I see in Table 3 increase of the mortality rate by 0.4% (not 0.8%) regarding PM2.5 and 0.3% regarding PM10 (not 0.6%). Both estimates are not statistically significant – please mention that in the result section!

In the abstract the values are stated as 0.8% regarding PM2.5 and 0.8% regarding PM10. Which values are correct (0.3, 0.6 or 0.8% increase of mortality rate regarding PM10)? This is a total mess.

  1. Discussion

“This study's results are consistent with those shown in the US, Italy, or China, showing that chronic exposure to air particles is associated with COVID-19 incidence and mortality” – this sentence is incorrect, as in this study the association between chronic exposure to PM2.5 and PM10 and COVID-19 mortality is not statistically significant.

Consequently, also the sentence in the abstract “Conclusions: The adjusted regression models showed that the COVID-19 incidence and mortality rate were significantly associated with chronic exposure to PM2.5 and PM10, even after adjusting by other variables” is not true. The COVID-19 mortality rate in this study is NOT significantly associated with chronic exposure to PM2.5 and PM10. Please change also the corresponding sentences in the conclusion section.

By the way, the authors do not discuss the fact that the estimates for the increase of the mortality rates associated with an increase of PM2.5 or PM10 in other studies were much larger than in this study (8% in the US study, 9% in the Italian study). In this study < 1% and not statistically significant. It should be clearly pointed out and the reasons for this should be discussed. First of all, similarities and differences between the studies regarding output data (health and exposure), study design, statistical analysis, others...

Author Response

Comment 1: “The COVID-19 confirmed cases and deaths were obtained at commune level from the website of the Chilean Ministry of Science” –there is no information how the COVOD-19 cases were confirmed, by PCR? ICD code? Please provide the information”

Answer: This information was included in subsection “2.2.1. Health variables” on page 6.

Comment 2: “The annual average concentration of PM2.5 and PM10 were estimated by a chemical transport model. It is not clear for which year the PM concentration were modeled. Please state it. How was the model validated? I guess that in Chile network monitoring stations a reoperated. It would be nice to see the comparison of the modeled and measured PM2.5 and PM10 values.”
Answer: We have added more details in the methodology section (subsection 2.2.2. Air pollution). Also, we add a new figure (figure1) to illustrate the estimations of our WRF-CHIMERE models with the reference measurements. We modeled year 2016, and we have given more details in the manuscript on pages 7-9.
Comment 3: “I don’t understand the sentence “For each commune, the annual average of PM2.5 and PM10 was estimated as the average of the daily concentrations estimated by the model for each grid point”: the annual average was estimated as the average of daily concentrations? Please clarify it.”
Answer: The annual average concentration for both pollutants was estimated as the average of daily concentrations for the year 2016. We have added more detail in the last paragraph on page 7.

Comment 4: “Temperature and humidity. The authors state that the average temperature and relative humidity were calculated based on data from 1980 and 2010. I agree that this isa long time period, however we know that we are currently facing a climate change and that the last few years are not necessarily comparable to the years before 2010. The last few years have been globally much warmer than the years before. How sure could the authors be that the year 2020 can be well represented by the mean values for the period 1980-2010?”
Answer: The average annual air temperature and relative humidity between 1980-2010 for each commune was obtained from ARClim, which is an open-source meteorological dataset. This dataset was chosen due to its official character given by the use by the Chilean Ministry of Environment, and because its spatial resolution (5x5km) allows to have data for each commune. We have added this statement and the reference on page 9. Unfortunately, we do not have available data regarding meteorological factors for all communes in this study for the 2020 year.

Comment 5: “Confounders. Self-report of health: this could be described in more detail. Which diseases are they, at least which groups of diseases are included? Movement index: please provide the definition of movement index”
Answer: The requested details were included on pages 9 and 10.

Comment 6: “Table 1: Obviously the results of the air pollution are available only for188 communes. What is the reason for it? I guess that the analysis was done only for the 188 communes with the PM2.5 and PM10modeled values. Please restrict all data (in tables and figures) only to those communes, or at least provide a second table with the health
and socio-environmental statistics only for n=188 (and not 346). The same is valid for Figure 1 – please show the spatial distribution of all health variables, temperature and humidity as well as of the confounders only for the communes included into the analysis.”

Answer: We updated table 1 and figure 2 (before it was figure 1). From now on, we only include information about 188 communes in all our tables and figures.

Comment 7: “Figure 2 and figure 3: please state the values of the quantiles for PM2.5 and PM10”

Answer: We have stated the values of the quantiles for PM2.5 as a figure caption in figure 3.

Comment 8: “Table 2 and 3: Please explain the abbreviations IRR and IC95%. Please show the values of the confidence intervals (I assume that“IC95%” means this) in brackets and separated by something (colon or dash), but not in two different columns (for example (1.008 -1.017)). Why are the statistically significant IRR values in Table 2 in bold, but this is not done in Table 3?”
Answer: We have corrected this issue in both tables.

Comment 9: “Adjusted models confirmed these associations, taking into account important confounders. For each microgram per cubic meter, the incidence rate increased by 1.3% regarding PM2.5 and 0.9%regarding PM10”: I see in Table 2 increase of the incidence rate by1.2% regarding PM2.5 (not 1.3%)?”
Answer: It was a mistake, and we have corrected it (page 13).

Comment 10: “For each microgram per cubic meter, the mortality rate increased by0.8% regarding PM2.5 and 0.6% regarding PM10 (Table 2 and 3)”: I see in Table 3 increase of the mortality rate by 0.4% (not 0.8%) regarding PM2.5 and 0.3% regarding PM10 (not 0.6%). Both estimates are not statistically significant – please mention that in the result section!
Answer: In the last paragraph on page 13, we mentioned that the association between chronic exposure to air pollutants and the COVID-19 mortality rate was not statistically significant. In the discussion section, we discussed it.

Comment 11: “In the abstract the values are stated as 0.8% regarding PM2.5 and 0.8% regarding PM10. Which values are correct (0.3, 0.6 or 0.8% increase of mortality rate regarding PM10)? This is a total mess”.
Answer: We have addressed this point and corrected the abstract.

Comment 12: “Discussion. This study's results are consistent with those shown in the US, Italy, or China, showing that chronic exposure to air particles is associated with COVID-19 incidence and mortality” – this sentence is incorrect, as in this study the association between chronic exposure to PM2.5 andPM10 and COVID-19 mortality is not statistically
significant.”.
Answer: We have corrected this statement in the first paragraph of Discussion’s section on page 14.

Comment 13: “Consequently, also the sentence in the abstract “Conclusions: The adjusted regression models showed that the COVID-19 incidence and mortality rate were significantly associated with chronic exposure to PM2.5 and PM10, even after adjusting by other variables” is not true. The COVID-19 mortality rate in this study is NOT significantly associated with chronic exposure to PM2.5 and PM10. Please change also
the corresponding sentences in the conclusion section”.

Answer: We have corrected this sentence in the abstract, and we also have clarified that the mortality rate was not associated with air pollution exposure.

Comment 14: “By the way, the authors do not discuss the fact that the estimates for the increase of the mortality rates associated with an increase ofPM2.5 or PM10 in other studies were much larger than in this study (8% in the US study, 9% in the Italian study). In this study < 1% and not statistically significant. It should be clearly pointed out and the
reasons for this should be discussed. First of all, similarities and differences between the studies regarding output data (health and exposure), study design, statistical analysis, others”.
Answer: We have discussed the potential differences between our results and other studies in the discussion section of this new version on page 14.

We also have improved the background, references, description of our methods and results, and conclusions to correct the errors noted by the referee.

Reviewer 3 Report

This ia a very interesting paper that follows recent events occuring due to the pandemic. However, the authors must further elaborate on the methodology they followed, on the results obtained (for instance they mention the model CHIMERE but they do not present any results), on the data groups used (for instance they mention particulate matter data but they do not present samples of daily measurements or an analysis of max-min values and occurence circumstances) and on the final outcome of this work (for instance this paper might provide the tools and methods to collect different layers of gridded data and perform advanced statistical analyses and results)

Author Response

Comment 1: “This is a very interesting paper that follows recent events occurring due to the pandemic. However, the authors must further elaborate on the methodology they followed, on the results obtained (for instance they mention the model CHIMERE but they do not present any results), on the data groups used (for instance they mention particulate matter data but they do not present samples of daily measurements or an analysis of max-min values and occurrence circumstances) and on the final outcome of this work (for instance this paper might provide the tools and methods to collect different layers of gridded data and perform advanced statistical analyses and results)”
Answer: We have added more details about air pollution on pages 7-9. Also, we have added figure 1 to compare the estimates of our WRF-CHIMERE models to the reference measurements. We also have improved the background, references, description of our methods and results, and conclusions to correct the errors noted by the referee. Additionally, a native English speaker did the moderate English editions required by the referee.

Round 2

Reviewer 2 Report

In my opinion the authors have implemented all of my comments well and the manuscript is improved substantially. Taking into consideration that there have been only few studies conducted in South America on this topic, and that the revised manuscript is much better in terms of both content and language, I would vote for accepting it. 

Author Response

Comment: “In my opinion the authors have implemented all of my comments well and the manuscript is improved substantially. Taking into consideration that there have been only few studies conducted in South America on this topic, and that the revised manuscript is much better in terms of both content and language, I would vote for accepting it.”
Answer: We appreciated this comment. We also have done some fine/minor spell check.